# WHEN KNOWLEDGE HURTS: ENRICHING DOMAIN KNOWLEDGE FOR CAUSAL SCIENTIFIC REASONING

## ABSTRACT

Science has long sought to uncover the principles governing discovery, leaving progress in fields like materials science slow and labor-intensive. While Large Language Models (LLMs) can accelerate progress by integrating domain knowledge, we reveal the existence of a critical failure mode known as ***contextual tunneling***, wherein naive knowledge integration causes LLMs to over-anchor on narrow retrieval paths while suppressing broader parametric reasoning. Through the evaluation in materials discovery, we demonstrate that naive knowledge graph augmentation degrades performance by 20–35% on key reasoning tasks compared to direct prompting. To address this challenge, we introduce ARIA (Autonomous Reasoning Intelligence for Atomics), a causal-aware framework featuring: (i) hierarchical reasoning that provides graceful degradation to knowledge graph sparsity, (ii) enhanced analogic transfer for robust reasoning, (iii) knowledge graph enrichment through online searching. Extensive experiments show that, while naive KG integration consistently underperforms baseline LLMs, ARIA not only recovers this loss but also provides interpretable causal explanations by tracing reasoning through the knowledge graph, enabling scientists to verify and trust its outputs. Our work demonstrates that external knowledge can inadvertently constrain reasoning and establishes a principled framework for robust KG–LLM integration in scientific discovery.

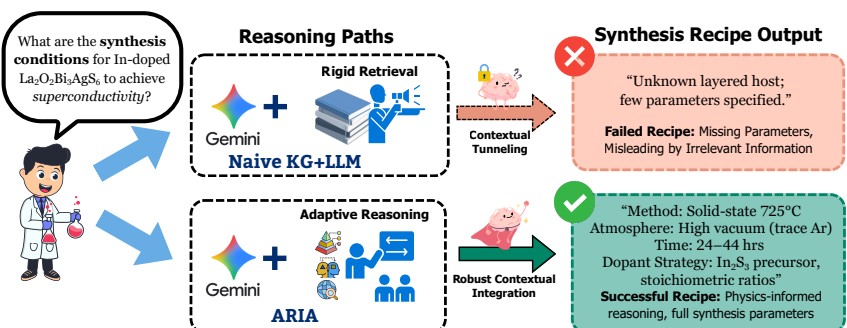

Figure 1: Naive LLM suffers from contextual tunneling issue (top) vs. our proposed ARIA with hierarchical reasoning (bottom). Our ARIA fundamentally overcomes the limitation of contextual tunneling and generate more accurate material parameters.

## 1 INTRODUCTION

While Large Language Models (LLMs) (Brown et al., 2020a) have demonstrated remarkable reasoning capabilities (Xu et al., 2025), their knowledge remains constrained by training data cutoffs and finite parametric capacity (Petroni et al., 2019; Brown et al., 2020b; Chowdhery et al.). These limitations often leads to factual inaccuracies and hallucinations (Li et al., 2024b), undermining their reliability for rigorous scientific inquiry. Retrieval-Augmented Generation (RAG) with Knowledge Graphs (KGs) has emerged as the standard solution, grounding LLMs in structured, domain-specific facts (Amayuelas et al., 2025; Liang et al., 2025; Edge et al., 2025). This approach has proven

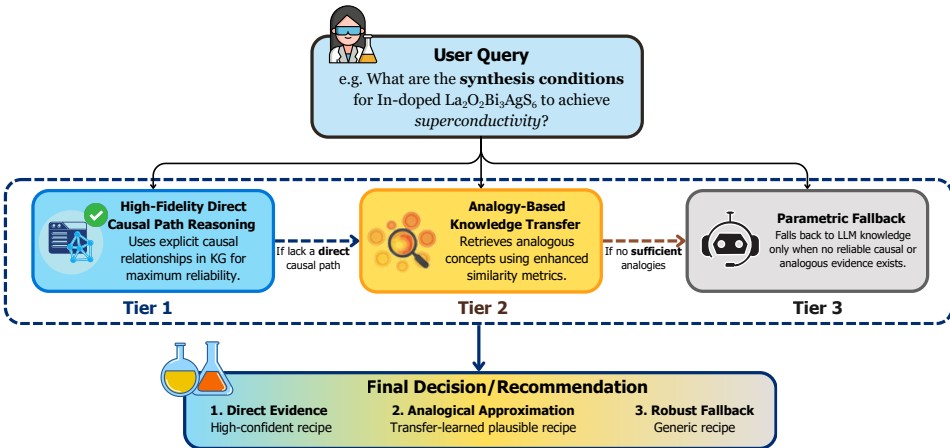

Figure 2: **Schematic of the** `ARIA` **Model Architecture.** The framework employs a three-tiered reasoning cascade. Tier 1 uses graph-constrained reasoning for queries with direct causal paths in the knowledge graph. If no such path exists, Tier 2 performs analogy-based reasoning by extrapolating from similar concepts. As a final step, Tier 3 provides a fallback mechanism, relying on the LLM's parametric knowledge when no external evidence is applicable. This hierarchical approach ensures outputs are maximally grounded in evidence while retaining flexibility to address novel queries.

successful for fact-based tasks such as question answering across chemistry, biology and materials science (Zhang et al., 2022; Wang et al., 2024; Bazgir et al., 2025).

Yet, as LLMs become more and more knowledgeable, recent studies challenge the assumption that external augmentation invariably improves reasoning (Yoran et al., 2024; Mallen et al., 2023). Inappropriate or incomplete retrieval can undermine rather than strengthen model performance (Wang et al., 2023; Xie et al., 2024). While prior work has identified this issue in commonsense settings, the prevailing response in specialized domains has been to "add more knowledge" (Zhang et al., 2021). We argue for a stronger conclusion: in complex scientific reasoning, irrelevant or narrow external knowledge can critically degrade performance.

To investigate this, we conducted a systematic evaluation in materials discovery, a domain that requires multi-step causal reasoning[1] over processing–structure–property relationships (Butler et al., 2018; Schmidt et al., 2019).

Our results reveal that naive KG integration leads to severe performance drops (20–35% compared to direct prompting) on both forward prediction of material properties and inverse design of synthesis protocols (Kim et al., 2020; Na, 2023). We attribute this failure to a core mechanism we term *Contextual Tunneling*: LLMs over-anchor on narrowly retrieved knowledge paths while suppressing their broader, more flexible parametric knowledge, as Figure 1 shows. We coin this in analogy to "cognitive tunneling" from psychology, where under stress individuals attend too narrowly to a single display (e.g., a pilot fixating on a head-up screen) while neglecting equally critical peripheral cues (Thomas & Wickens, 2001; Jarmasz et al., 2005).

To address this fundamental challenge, we introduce `ARIA` (Autonomous Reasoning Intelligence for Atomics), a causally-aware framework that enables selective and effective knowledge utilization. Instead of blindly injecting retrieved text, `ARIA` mitigates contextual tunneling through three synergistic endeavors: (1) **Hierarchical reasoning**, which adapts a three-tiered reasoning cascade, enabling graceful degradation when specific causal paths are absent and preventing over-reliance on narrow retrieval; (2) **Transfer learning**, which leverages similarity-based analogy to adapt causal relations to novel contexts while preserving mechanistic fidelity; and (3) **Dynamic KG enrichment**, which augments the knowledge base with information retrieved via web search, followed by a post-

---

[1]In this work, we define "causality" in the mechanistic sense established by the materials science Processing–Structure–Property (PSP) paradigm, where synthesis conditions physically determine resulting structure. This is distinct from statistical causal discovery approaches (e.g., PC algorithm) used in tabular data, as our Causal Knowledge Graph encodes verified physical mechanisms extracted from the literature.

hoc filtering stage to ensure high quality. We benchmark `ARIA` against the Baseline LLM, Naive KG+LLM, Online KG+LLM. Notably, naive KG integration degrades performance, In contrast, `ARIA` consistently rescues KG–LLM integration, achieving robust causal reasoning across tasks of varying difficulty.

Our key takeaway for practitioners is that *sometimes knowledge can hurt*: external knowledge may inadvertently constrain reasoning and reduce generalization. By diagnosing and addressing contextual tunneling, `ARIA` establishes a principled framework for robust, generalizable KG–LLM integration, advancing AI for scientific discovery and beyond.

## 2 RELATED WORK

**Knowledge-augmented generation** enhances LLMs with external knowledge to improve factual grounding (Lewis et al., 2020; Li et al., 2024a). This is especially required in rigorous science, medical, law and other domain specific reasoning scenarios (Zhang et al., 2022; Wang et al., 2024; Hou et al., 2025), where LLMs tend to hallucinate and make up misleading facts (Huang et al., 2025). Integrating causal knowledge graphs provides a more interpretable and reliable output by modeling underlying inference (Zhang et al., 2024; Samarajeewa et al., 2024). However, recent studies show that retrieving irrelevant information can create knowledge conflicts, preventing the model from utilizing its own parametric knowledge (Longpre et al., 2021; Xu et al., 2024). Related to our findings, GIVE (He et al., 2024) proposes a training-free reasoning framework that guides LLMs to merge parametric and non-parametric memories while mitigating noise in large or incomplete knowledge sources, highlighting a broader need to control retrieval-induced reasoning failures. Our work demonstrates that this failure mode extends to specialized scientific domains.

**The application of LLMs to materials science** has emerged as a promising avenue for accelerating discovery, with demonstrated capabilities in property prediction and synthesis planning (Zheng et al., 2023; D. White et al., 2023; Dagdelen et al., 2024). Early approaches primarily relied on fine-tuning domain-specific corpora to capture materials knowledge (Gupta et al., 2022; Jiang et al., 2025), while more recent work has explored prompt engineering and in-context learning for scientific reasoning (Jiang et al., 2025). Several systems have further integrated structured knowledge with LLMs. For example, MatChat (Chen et al., 2023) and AtomGPT (Choudhary, 2024) couples databases with conversational interfaces, ChemCrow demonstrates LLM-assisted synthesis planning (Bran et al., 2023).

Recent efforts have begun addressing these issues through causal reasoning (Zhang et al., 2024) and multi-modal integration (Samarajeewa et al., 2024). Yet, comprehensive frameworks that jointly enhance reasoning transparency, broaden contextual grounding, and enable transferable synthesis remain lacking. Our work advances this direction by introducing hierarchical reasoning, dyanamic KGs enrichment, and transferable synthesis for robust materials discovery.

## 3 METHOD

In this section, we introduce `ARIA`, a framework designed to enhance the reliability of scientific reasoning in LLMs. Our approach is motivated by a critical failure mode in retrieval-augmented systems, where irrelevant context degrades performance. We term this problem ***Contextual Tunneling*** and provide a formal definition in subsection 3.1. Next, in subsection 3.2, we detail the automated pipeline for constructing the Causal Knowledge Graph that serves as the evidentiary backbone for our system. With this foundation, we present the core architecture of `ARIA` in subsection 3.3: a principled, three-tiered reasoning engine that intelligently navigates between graph-based evidence, analogical inference, and the LLM's parametric knowledge. Finally, we ground our method in subsection 3.4 by formalizing the high-impact materials design tasks used to validate our approach.

### 3.1 CONTEXTUAL TUNNELING

Standard RAG pipelines enhance a large language model $f_{\text{LLM}}$ by conditioning its output $\mathbf{y}$ on both a query $\mathbf{q}$ and a set of retrieved documents $\mathcal{C}_{\text{retrieved}}$. The objective is typically to maximize the conditional probability $p(\mathbf{y}|\mathbf{q}, \mathcal{C}_{\text{retrieved}})$. However, we identify a critical failure mode we term

*Contextual Tunneling*, where the model's performance degrades because it is forced to reason over an irrelevant, incomplete, or misleading context.

Formally, we define Contextual Tunneling as the phenomenon where the introduction of retrieved context $\mathcal{C}$ increases the divergence between the model's reasoning path and the optimal reasoning path. This can be quantified as a degradation in the Kullback-Leibler (KL) divergence:

$$D_{KL}(P(\mathbf{y}|\mathbf{q})||P(\mathbf{y}|\mathbf{q}, \mathcal{C}_{\text{narrow}})) > \epsilon \tag{1}$$

where the retrieved context $\mathcal{C}_{\text{narrow}}$ causes the attention mechanism to over-anchor on high-similarity but functionally irrelevant tokens, suppressing the activation of broader parametric knowledge.

This occurs when the retrieved context, $\mathcal{C}_{\text{narrow}}$ forces the model to anchor on a irrelevant reasoning path, resulting in a lower-quality output than relying on its parametric knowledge alone (Yu et al., 2024; Liu et al., 2024). We formalize this degradation as follows:

$$\mathbb{E}[\text{Quality}(f_{\text{LLM}}(\mathbf{q}, \mathcal{C}_{\text{narrow}}))] \leq \mathbb{E}[\text{Quality}(f_{\text{LLM}}(\mathbf{q}))], \tag{2}$$

where Quality(.) is any task-specific evaluation metric. Our work introduces a framework designed to explicitly prevent this negative contribution, ensuring that external knowledge serves as a reliable enhancement.

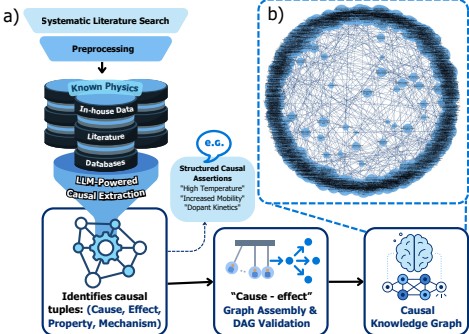

Figure 3: **Overview of the automated knowledge graph construction pipeline and its application to materials design.** (a) Workflow for constructing KGs from scientific literature. (b) Visualization of the resulting knowledge graph structure.

### 3.2 CAUSAL KNOWLEDGE GRAPH CONSTRUCTION

To ground ARIA's reasoning in verifiable domain knowledge, we construct a Causal Knowledge Graph by developing an automated pipeline that ingests a large corpus of scientific literature. This process, illustrated in Figure 3 (a), ensures the knowledge base is structured, attributable and scalable. Our pipeline comprises four stages: (1) Corpus acquisition and preprocessing, (2) LLM-Powered information extraction, (3) Dynamic knowledge enrichment, and (4) Final graph assembly.

The pipeline begins with a systematic scientific literature search, followed by domain-specific parsing and data cleaning. During preprocessing, we normalize scientific units (e.g., converting all temperatures to Kelvin and energies to electronvolts) and apply consistency checks such as valency- and stoichiometry-based filtering to eliminate chemically impossible or physically incoherent statements. For information extraction, we employ an LLM that is constrained by a predefined ontology governing allowed entity types, relation types, and numeric attributes. The model is required to output JSON objects that strictly follow this schema, ensuring structured and machine-verifiable extraction. Each resulting tuple $\mathcal{T}_1, \mathcal{T}_2, \ldots, \mathcal{T}_n$ encodes a (`cause`, `effect`, `relationship type`, `supporting text`) record.

To address the sparsity inherent in domain-specific knowledge graphs, we introduce a dynamic enrichment step (Rezayi et al., 2021). Here, the LLM is augmented with a web search tool to identify missing links, obtain parameter ranges, or retrieve corroborating evidence. All retrieved candidates are subjected to post-hoc validation—ensuring numeric coherence, removing contradictory relations, and verifying that evidence snippets directly support the extracted causal claim.

After enrichment, a quality-control filter prunes incomplete, underspecified, or weakly supported relations. The remaining tuples are compiled into a directed graph $\mathcal{G} = (\mathcal{V}, \mathcal{E})$, shown in Figure 3(b). Each unique `cause` or `effect` entity becomes a node in $\mathcal{V}$, and each tuple $\mathcal{T}_i$ generates a directed edge from the `cause` to the `effect`. Edge attributes store the relationship type, numerical metadata, and supporting evidence text, providing the rich contextual grounding that `ARIA` later exploits for mechanistic interpretation and provenance-aware reasoning (Liang et al., 2025; Bai et al., 2025).

### 3.3 `ARIA`: Autonomous Reasoning Intelligence for Atomics

As illustrated in Figure 2, the `ARIA` framework is designed to mitigate contextual tunneling by structuring the interaction between an LLM and a Causal Knowledge Graph through a principled, three-tiered reasoning cascade. This architecture emulates a rigorous scientific reasoning process: it prioritizes high-fidelity, direct evidence first, then resorts to principled analogical reasoning for novel problems, and finally relies on the LLM's general parametric knowledge only as a last resort

**Tier 1: high-fidelity direct causal path reasoning.** For queries where the core entities are well-represented in our causal knowledge graph, `ARIA` employs a graph-constrained reasoning approach. This tier prioritizes verifiable, explicit causal links to ensure the highest reliability. It first grounds the query's concepts onto the causal graph, then traverses its structure to elicit all verifiable causal pathways connecting them. (Jin et al., 2024) This extracted evidence then serves as a symbolic scaffold that directly constrains the LLM's generation (DeLong et al., 2025), producing a high-fidelity output that faithfully reflects the corpus.

**Tier 2: analogy-based knowledge transfer.** If a direct causal path is unavailable, often the case for novel or out-of-distribution query, `ARIA` switches to the second tier: analogy-based approach.

This approach retrieves a set of the most relevant analogous concepts, denoted $\mathcal{V}_{\text{analogous}}$, from the knowledge graph. The retrieval is a two-stage process. First, we identify a set of all plausible candidates, $\mathcal{V}_{\text{plausible}}$, by filtering for nodes whose similarity score exceeds a predefined threshold $\tau$:

$$\mathcal{V}_{\text{plausible}} = v \in \mathcal{V} \mid \text{Sim}_{\text{enhanced}}(\mathbf{q}, v) \geq \tau. \tag{3}$$

From this set, we select the final top-K nodes with the highest similarity scores to form our context:

$$\mathcal{V}_{\text{analogous}} = \underset{v \in \mathcal{V}_{\text{plausible}}}{\text{Top-K}} \left( \text{Sim}_{\text{enhanced}}(\mathbf{q}, v) \right). \tag{4}$$

To ensure analogies remain physically meaningful in scientific domains—where surface-level semantic similarity is insufficient—we augment the similarity function to incorporate factual and numerical plausibility:

$$\text{Sim}_{\text{enhanced}}(\mathbf{q}, v) = w_1 \cdot \cos(\mathbf{h}_{\mathbf{q}}, \mathbf{h}_v) + w_2 \cdot \text{FC}(\mathbf{q}, v) + w_3 \cdot \text{NC}(\mathbf{q}, v). \tag{5}$$

**Factual Consistency (FC).** We formalize FC as a binary categorical mask that enforces ontology-level compatibility:

$$\text{FC}(\mathbf{q}, v) = \mathbb{1}_{\text{cat}}(\mathbf{q}, v), \tag{6}$$

where $\mathbb{1}_{\text{cat}}(\mathbf{q}, v) = 1$ if the query and candidate belong to the same material category (e.g., both p-type semiconductors, both chalcogenides), and 0 otherwise. This prevents analogies that are semantically plausible but categorically contradictory.

**Numerical Compatibility (NC).** To quantify physical compatibility of continuous parameters (e.g., temperature, energy, pressure), we compute:

$$\text{NC}(\mathbf{q}, v) = \exp\!\left( -\frac{\|x_q - \mu_v\|^2}{2\sigma^2} \right), \tag{7}$$

where $x_q$ is the query's numerical attribute (such as required annealing temperature), $\mu_v$ is the candidate node's valid-range mean, and $\sigma$ controls the sensitivity to deviations. This penalizes nodes that may be semantically similar but violate physical constraints (e.g., incompatible melting points or stability windows).

The causal pathways associated with the nodes in $\mathcal{V}_{\text{analogous}}$ are then aggregated and used as templates to construct a hypothesis for the original query.

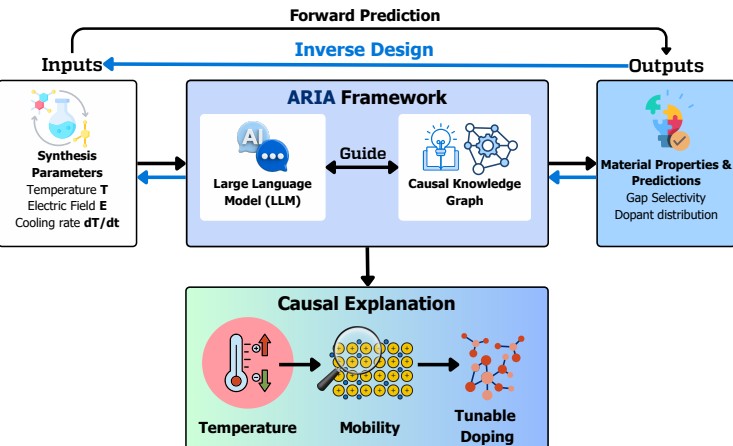

Figure 4: **Schematic of the `ARIA` framework for bidirectional reasoning in materials discovery.** The framework predicts material properties from synthesis parameters in forward tasks, while enabling inverse design by generating synthesis protocols from target properties.

**Tier 3: Parametric fallback.** If the Causal Knowledge Graph contains no direct path and no sufficiently analogous concepts (i.e., the highest $\text{Sim}_{\text{enhanced}}$ score is below a predefined threshold $\tau$), `ARIA` defaults to its third tier. In this mode, it forgoes the external knowledge and prompts the LLM directly, relying solely on the model's parametric knowledge. This prevents contextual tunneling by avoiding the use of low-quality or irrelevant retrieved information.

**Tier selection.** ARIA's final output is generated by a cascading selection mechanism based on the availability of evidence in the Causal Knowledge Graph $\mathcal{G}$. The framework evaluates the tiers sequentially: it first attempts to find direct causal evidence (Tier 1). If no direct path exists, it then searches for sufficiently similar analogous evidences (Tier 2). If neither form of evidence is found, the system defaults to using its internal parametric knowledge (Tier 3).

This selection logic for a given query $q$ is formalized as follows:

$$\text{ARIA} = \begin{cases} f_{\text{direct}}(q, \mathcal{P}_{\text{direct}}) & \text{if exact path exists in graph } \mathcal{G} \\ f_{\text{transfer}}(q, \mathcal{P}_{\text{analogue}}) & \text{if } \mathcal{P}_{\text{direct}} = \emptyset \text{ and } \text{Sim}_{\text{enhanced}}(q, v^*) \geq \tau \\ f_{\text{parametric}}(q) & \text{Otherwise,} \end{cases} \quad (8)$$

where $\mathcal{P}_{\text{direct}}$ is the set of direct causal paths retrieved for Tier 1 and $\mathcal{P}_{\text{analogue}}$ is the set of causal paths constructed from analogous nodes. $f_{\text{direct}}$, $f_{\text{analogy}}$ and $f_{\text{parametric}}$ are generation functions for each respective tier. This architecture grounds outputs in evidence when possible, while retaining the flexibility to reason about novel challenges in a controlled and transparent manner.

## 3.4 MATERIALS DESIGN TASKS

The `ARIA` framework is designed to solve complex causal reasoning problems, which we formalize here using a high-impact application: the central challenges of materials discovery. This domain is governed by the foundational processing-structure-property (PSP) paradigm of materials science (Butler et al., 2018; Schmidt et al., 2019). As illustrate in Figure 4, this paradigm posits that the manufacturing **process** ($\mathcal{S}$) causally determines a material's internal **structure** ($\mathcal{M}$), which in turn dictates its functional **properties** ($\mathcal{P}$). Our tasks are to reason across this complex, multi-scale causal chain.

**Forward prediction: from process to properties.** The forward problem mirrors the task of predicting the outcome of a novel experiment. Given a set of synthesis conditions $\mathcal{S}$ (e.g., precursor chemicals, temperature, pressure), the goal is to predict the final material properties $\mathcal{P}$ (e.g., conductivity, bandgap, stability). This is a cascaded function where synthesis determines structure, and

structure determines properties:

$$\hat{\mathcal{P}} = f(\mathcal{S}) = g(h(\mathcal{S})) \tag{9}$$

Here, $h : \mathcal{S} \rightarrow \mathcal{M}$ maps synthesis to structure (e.g., crystal phase, grain size), and $g : \mathcal{M} \rightarrow \mathcal{P}$ maps that structure to its resulting properties.

**Inverse design: from properties to process.** The inverse problem represents the "holy grail" of materials discovery: given a set of target properties $\mathcal{P}^*$, the goal is to identify an optimal set of processing conditions $\mathcal{S}^*$ to synthesize the desired material. This is a far more challenging task, as it requires searching a vast and highly constrained space of possible synthesis recipes $\Omega$:

$$\mathcal{S}^* = \arg\min_{\mathcal{S} \in \Omega} \|\mathcal{P}^* - f(\mathcal{S})\|^2 + \lambda R(\mathcal{S}). \tag{10}$$

The regularization term $R(\mathcal{S})$ is crucial as it constrains the search to physically realizable and experimentally viable synthesis protocols, avoiding impossible or impractical solutions. For example, a typical task is to find the precise chemical vapor deposition (CVD) conditions required to grow a 2D material with a target electronic bandgap.

## 4 EXPERIMENTS

### 4.1 EXPERIMENTAL SETUP

**Datasets and knowledge graph.** We constructed a dataset with knowledge graph for our materials design evaluation from peer-reviewed literature. Using the method from subsection 3.2, the knowledge graph comprises 149 synthesis-property relationships across 85 distinct materials systems (e.g., semiconductors, superconductors, 2D materials). The dataset is partitioned into an **in-domain** set of 117 experiments and a challenging **out-of-domain** set designed to test generalization on novel materials. Each entry contains expert-validated ground truth for synthesis conditions, structural changes, and property outcomes, enriched with mechanistic explanations. This design enables a rigorous evaluation of both in-distribution performance and the model's ability to generalize its causal reasoning, mirroring real-world scientific discovery challenges.

**Baselines.** We evaluate `ARIA` against a diverse set of baselines to ensure a comprehensive comparison. These baselines are: 1) **Baseline LLM**: The base *gemini-1.5-pro-latest* model without any external knowledge augmentation, relies solely on its pre-trained knowledge, isolating the impact of any retrieval-based method. 2) **Naive KG+LLM**: A conventional RAG implementation that retrieves context from our Causal Knowledge Graph via cosine similarity, but lacks `ARIA`'s tiered reasoning and fallback mechanisms. 3) **Online KG+LLM**: A RAG baseline that utilizes a live online search tool in addition to the curated knowledge graph, grounding its responses with dynamic, real-time information.

**Evaluation framework.** To ensure scientific validity, we employ *gemini-1.5-pro-latest* as an expert LLM judge to evaluate both the final prediction and its supporting explanation (Team et al., 2024). Following a detailed rubric, each model output is scored from 0-10 across a multi-dimensional set of criteria. This multi-dimensional evaluation assesses correctness via **scientific accuracy** (adherence to physical principles) and **functional equivalence** (achieving the target outcome), as well as the explanation's quality through its **reasoning quality** (logical coherence), **completeness** and **interpretability**. A final **overall score** provides a holistic assessment of practical utility. This LLM-judge approach is essential for capturing the domain-specific nuance required to evaluate complex scientific reasoning, a known limitation of traditional automated metrics.

**Implementation details.** All experiments are conducted using *gemini-1.5-pro-latest* as the base large language model. For all retrieval and similarity-based reasoning tasks, we generate embeddings using the *all-MiniLM-L6-v2* model. A cosine similarity threshold of 0.6 is used for node retrieval in our Causal Knowledge Graph. Complete details on our prompt engineering strategies and evaluation rubrics are provided in Appendix A.

## 4.2 EMPIRICAL VALIDATION OF CONTEXTUAL TUNNELING: CASE STUDY

We evaluate the framework on a challenging inverse design task that reveals contextual tunneling in naive KG-LLM approaches. In our case study (see subsection B.2 and Figure 1), the naive model became fixated on irrelevant analogies and produced vague "intercalation or alloying" recommendations without any concrete synthesis parameters (Table 3). In contrast, ARIA maintains broad contextual reasoning, providing detailed protocols including specific temperature ranges (800-1200°C), controlled atmospheres, and systematic characterization steps. This demonstrates how causally-grounded frameworks prevent tunnel vision by preserving reasoning capabilities across material properties and synthesis requirements, with detailed analysis in the subsection B.1 and subsection B.2.

Table 1: **In-domain vs. out-of-domain performance analysis.** We evaluate four systems on in-domain data (materials/protocols covered in KG) and out-of-domain data (novel materials/protocols not in KG). ARIA demonstrates superior generalization across both forward prediction and inverse design tasks, rescuing performance degradation from naive KG integration.

| System | Domain | Scientific Accuracy | Functional Equivalence | Reasoning Quality | Completeness | Interpretability | Overall |
|---|---|---|---|---|---|---|---|
| | | **Forward Prediction** | | | | | |
| Baseline LLM | In-Domain | 0.68 | 0.42 | 0.66 | 0.33 | 0.68 | 0.52 |
| Baseline LLM | Out-of-Domain | 0.65 | 0.38 | 0.61 | 0.29 | 0.62 | 0.47 |
| *Domain Gap* | | *-5.4%* | *-11.0%* | *-7.5%* | *-10.5%* | *-8.7%* | *-10.0%* |
| Naive KG+LLM | In-Domain | 0.48 | 0.29 | 0.42 | 0.20 | 0.46 | 0.34 |
| Naive KG+LLM | Out-of-Domain | 0.49 | 0.29 | 0.45 | 0.22 | 0.50 | 0.37 |
| *Domain Gap* | | *+1.2%* | *+1.5%* | *+8.2%* | *+9.2%* | *+7.5%* | *+6.4%* |
| Online KG+LLM | In-Domain | 0.62 | 0.35 | 0.57 | 0.25 | 0.61 | 0.43 |
| Online KG+LLM | Out-of-Domain | 0.64 | 0.38 | 0.62 | 0.27 | 0.65 | 0.46 |
| *Domain Gap* | | *+3.8%* | *+6.6%* | *+7.7%* | *+8.2%* | *+6.5%* | *+6.5%* |
| ARIA | In-Domain | 0.62 | 0.36 | 0.58 | 0.25 | 0.61 | 0.44 |
| ARIA | Out-of-Domain | 0.61 | 0.33 | 0.57 | 0.23 | 0.60 | 0.42 |
| *Domain Gap* | | *-1.9%* | *-6.6%* | *-0.7%* | *-8.0%* | *-1.0%* | *-4.2%* |
| | | **Performance Comparison** | | | | | |
| Naive KG+LLM vs Baseline | | *-24.2%* | *-22.5%* | *-25.3%* | *-26.6%* | *-20.0%* | *-21.5%* |
| Online KG+LLM vs Baseline | | *-1.0%* | *+0.8%* | *+2.1%* | *-7.4%* | *+4.0%* | *-0.7%* |
| ARIA vs Baseline | | *-6.3%* | *-10.8%* | *-5.7%* | *-21.3%* | *-3.5%* | *-9.4%* |
| ARIA vs Naive KG | | *+23.6%* | *+15.1%* | *+26.2%* | *+7.2%* | *+20.6%* | *+15.4%* |
| | | **Inverse Design** | | | | | |
| Baseline LLM | In-Domain | 0.62 | 0.48 | 0.60 | 0.50 | 0.66 | 0.56 |
| Baseline LLM | Out-of-Domain | 0.64 | 0.47 | 0.64 | 0.53 | 0.71 | 0.59 |
| *Domain Gap* | | *+3.8%* | *-3.2%* | *+7.0%* | *+6.7%* | *+6.9%* | *+4.0%* |
| Naive KG+LLM | In-Domain | 0.46 | 0.37 | 0.39 | 0.40 | 0.46 | 0.41 |
| Naive KG+LLM | Out-of-Domain | 0.47 | 0.35 | 0.41 | 0.40 | 0.49 | 0.42 |
| *Domain Gap* | | *+2.7%* | *-3.5%* | *+6.1%* | *+0.2%* | *+5.5%* | *+1.6%* |
| Online KG+LLM | In-Domain | 0.61 | 0.48 | 0.56 | 0.50 | 0.63 | 0.54 |
| Online KG+LLM | Out-of-Domain | 0.57 | 0.42 | 0.50 | 0.49 | 0.59 | 0.50 |
| *Domain Gap* | | *-6.0%* | *-12.7%* | *-9.7%* | *-2.5%* | *-5.1%* | *-6.4%* |
| ARIA | In-Domain | 0.58 | 0.44 | 0.55 | 0.47 | 0.63 | 0.52 |
| ARIA | Out-of-Domain | 0.63 | 0.47 | 0.59 | 0.53 | 0.67 | 0.57 |
| *Domain Gap* | | *+9.6%* | *+8.1%* | *+8.4%* | *+11.3%* | *+6.8%* | *+9.7%* |
| | | **Performance Comparison** | | | | | |
| Naive KG+LLM vs Baseline | | *-25.9%* | *-24.2%* | *-35.3%* | *-25.1%* | *-30.5%* | *-28.7%* |
| Online KG+LLM vs Baseline | | *-11.2%* | *-10.7%* | *-21.1%* | *-8.2%* | *-15.9%* | *-14.4%* |
| ARIA vs Baseline | | *-1.5%* | *+2.0%* | *-6.9%* | *-1.2%* | *-4.9%* | *-2.7%* |
| ARIA vs Naive KG | | *+32.9%* | *+34.5%* | *+43.9%* | *+32.0%* | *+36.9%* | *+36.6%* |

## 4.3 MAIN RESULTS

We evaluate four systems, Baseline LLM, Naive KG+LLM, Online KG+LLM and ARIA across in-domain and out-of-domain datasets, to assess how knowledge graph integration affects reasoning. Table 1 and Figure 5 show the performance across six metrics for two material discovery tasks:

forward prediction and inverse design. Overall, we observe that `ARIA` presents to be a powerful method against contextual tunneling. We discuss particular observations below:

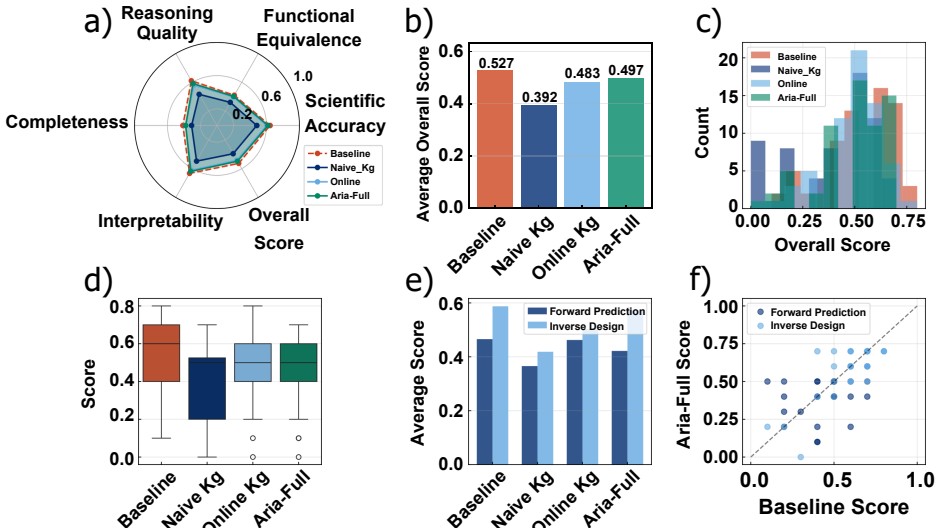

Figure 5: **Comprehensive evaluation framework comparing baseline LLM, KG-LLM, Online-LLM and ARIA performance on materials science tasks using LLM-based scoring.** (a) Multi-dimensional performance profile showing radar plot comparison of average scores across six evaluation criteria. (b) Overall model performance comparison showing average overall scores across all tasks and datasets. (c) Distribution of performance improvements relative to baseline across forward prediction and inverse design tasks. (d) Box plot comparison of overall score distributions across all three models, showing median, quartiles, and outliers. (e) Performance breakdown by task type (forward prediction vs. inverse design) for all models, illustrating task-specific strengths and weaknesses. (f) Head-to-head scatter plot comparison between the best-performing structured model and baseline, with each point representing one test case. Points below the diagonal line indicate baseline superiority.

**Naive knowledge integration triggers contextual tunneling.** While the case study in subsection 4.2 provides an empirical view of contextual tunneling, our experiments demonstrate that this is not an isolated example but a systematic issue. We find that naively integrating the knowledge graph (Naive KG+LLM) is actively harmful. This approach consistently underperforms the parametric-only Baseline LLM, with overall performance degrading by over 28.7% in complex, out-of-domain inverse design tasks. Interestingly, Naive KG+LLM performs slightly less degradation in the out-of-domain than in-domain task. This indicates that naive integration or simple knowledge injection introduces noise and potentially over-conditioning during the generation, undermining analytical capability rather than facilitating knowledge transfer.

**The limits of online searching.** An analysis of the Online KG+LLM baseline reveals a critical insight: simply providing more, even real-time, information is not a universal solution. For the forward prediction task, the online search is highly effective, achieving performance nearly identical to the Baseline LLM (-0.7% overall). However, for the more complex inverse design task requiring multi-step causal reasoning, online searching causes a significant performance degradation of -14.4%. This asymmetry demonstrates a more subtle form of Contextual Tunneling: while web search can retrieve abundant factual evidence, it does not inherently enhance the model's ability to synthesize a coherent, multi-step plan.

**ARIA demonstrates a powerful "rescue effect".** ARIA successfully reverses the performance degradation caused by naive graph integration. This "rescue effect" is most pronounced in challenging, out-of-domain scenarios, where `ARIA` improves upon the Naive KG+LLM method by a substantial 36.6% in inverse design tasks, restoring performance to near-baseline levels. While

also significant in forward prediction (up to 21.5% improvement), the amplified gains in the more complex inverse design setting underscore ARIA's strength in multi-step reasoning. Furthermore, `ARIA` enhances domain generalization; for instance, it shrinks the performance gap on the out-of-domain forward prediction task from -10% (in the Baseline LLM) to just -4.2%, transforming external knowledge from a source of interference into a tangible asset for generalization.

**ARIA enhances both reasoning quality and interpretability.** A metric-specific breakdown shows that `ARIA`'s largest gains occur in the structure and clarity of the generated reasoning. In the inverse design task, Reasoning Quality increases by nearly 44% relative to the naïve method, accompanied by a 37% improvement in interpretability. These advances indicate that `ARIA` not only improves correctness but also produces more logically coherent and human-readable explanations—an essential attribute in scientific reasoning, where explanatory rigor is as important as predictive accuracy.

**The trade-off between accuracy and provenance.** While the Baseline LLM achieves high numerical scores, it functions as a black box: its answers lack citations, verifiable grounding, and explicit evidence trails. In scientific discovery, such provenance is essential. Naive RAG introduces provenance but often sacrifices accuracy due to contextual tunneling. `ARIA` resolves this tension by offering a "glass box" alternative—recovering the strong performance of the Baseline LLM while grounding each reasoning step in the Causal Knowledge Graph. This achieves the dual goals of high predictive accuracy and scientifically interpretable, fully traceable reasoning.

## 5 LIMITATIONS AND FUTURE WORKS

**Task complexity considerations.** Our evaluation does not distinguish simple tasks solvable with parametric knowledge from complex ones requiring deeper causal reasoning. Future work should stratify tasks to better expose when contextual tunneling arises.

**Evaluation framework limitations.** Using an LLM judge risks bias toward fluent but less structured outputs, potentially obscuring `ARIA`'s strengths in verifiability. Expert or human-in-the-loop review could offer more faithful evaluation.

**Towards more reliable, transparent and autonomous scientific reasoning.** Grounding on narrow knowledge bases limits discovery. Progress demands agentic frameworks that synthesize evidence across diverse, multimodal sources—moving beyond RAG toward autonomous scientific reasoning.

## 6 CONCLUSION

In this work, we identified *Contextual Tunneling*, a critical failure mode where naive knowledge augmentation degrades an LLM's scientific reasoning. We introduce `ARIA`, a framework that mitigates this issue with a tiered reasoning cascade for selective knowledge integration. Experiments in materials science discovery confirm `ARIA` recovers the performance loss from naive RAG, demonstrating that the method of integration is as critical as the knowledge itself. Ultimately, `ARIA` provides a principled approach for robust and interpretable KG-LLM integration, advancing the development of reliable AI for scientific discovery.

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

## A  PROMPT TEMPLATES

### A.1  HIGH-FIDELITY DIRECT CAUSAL PATH REASONING

```
You are an expert materials scientist with access to a specialized
    knowledge graph derived from 200+ research papers.
Your task is to {task_desc} by intelligently combining your baseline
    scientific knowledge with relevant research findings.

**CRITICAL INSTRUCTION: Your final answer must be AT LEAST as good as
    pure baseline reasoning. Use DAG knowledge to ENHANCE, not replace,
    fundamental principles.**

**{input_label}:**
{json.dumps(original_prompt_data, indent=2)}

**Relevant Research Knowledge from Literature:**
Causal Pathways:
- {formatted_paths}

Known Mechanisms:
- {formatted_mechanisms}

{similarity_context}

**Integration Strategy ({quality_assessment['recommendation']}):**
1. **Baseline Analysis**: First, provide your fundamental materials
    science analysis
2. **DAG Enhancement**: Use the research knowledge to enhance or validate
     your baseline reasoning
3. **Quality Control**: Ensure the final prediction is scientifically
    sound and improves upon baseline
4. **Confidence Assessment**: Provide honest confidence levels for each
    aspect

**Output Instructions:**

Your response should follow this two-part structure:
**Part 1: Step-by-Step Reasoning**
First, write out your detailed thought process as plain text. Follow the
    integration strategy below:
1. **Baseline Analysis**: Provide your fundamental materials science
    analysis based on the inputs.
2. **DAG Enhancement**: Use the provided research knowledge to enhance,
    validate, or refine your baseline reasoning.
3. **Synthesis & Conclusion**: Combine both knowledge sources to form a
    final, scientifically rigorous conclusion. Explain the mechanisms
    involved.

**Part 2: Final JSON Output**
After you have written your reasoning, provide the final answer as a
    single, valid JSON object inside a JSON code block. The 'reasoning'
    key within the JSON should be a concise summary of your detailed
    reasoning from Part 1.

**JSON Rules (for Part 2):**
1. The JSON code block **MUST** contain a single, valid, RFC 8259
    compliant JSON object.
2. Comments are strictly forbidden inside the JSON.
3. All keys and all string values **MUST** be enclosed in double quotes.
4. No trailing commas are allowed.

**JSON Output Format:**
{output_format}
```

## A.2 TRANSFER LEARNING

```
You are an expert materials scientist AI conducting transfer learning
    analysis. Your knowledge graph lacks exact pathways, but you've
    identified analogous information that requires careful validation and
     adaptation.

**Task:**
Based on analogous information and comprehensive literature search, {
    task_description} for the user's target.

**{input_data_label} (User's Query):**
{json.dumps(original_prompt_data, indent=2)}

**Similar Known Causal Pathways:**
- {formatted_paths}

**Known Mechanisms for Similar Pathway:**
- {formatted_mechanisms}

**Similarity Analysis:**
- Embedding distance: {property_embedding_diff:.4f} (0=identical, 2=
    opposite)
- Most similar known case: {similar_node}

Your response should follow this two-part structure:

**Part 1: Step-by-Step Reasoning**
1. **Analyze & Compare:** Briefly compare the User's Query with the Known
     Pathway. What are the key similarities and, more importantly, the
    key differences (e.g., opposite doping type, different materials,
    different conditions)?
2. **Formulate Hypothesis:** Based on the differences and the
    quantitative embedding distance, state a hypothesis.
3. **Extrapolate or Diverge:** Decide if you can adjust the parameters
    from the known pathway (extrapolate) or if you must suggest a
    completely different approach (diverge). Justify this decision using
    the embedding distance. A small distance (< 0.4) suggests
    extrapolation is viable; a large distance (> 0.7) suggests divergence
     is necessary.
4. **Synthesize Final Answer:** Based on your hypothesis, construct the
    final prediction/suggestion.

**Part 2: Final JSON Output**
After you have written your reasoning, provide the final answer as a
    single, valid JSON object inside a JSON code block. The 'reasoning'
    key within the JSON should be a concise summary of your detailed
    reasoning from Part 1.

**JSON Rules (for Part 2):**
1. The JSON code block **MUST** contain a single, valid, RFC 8259
    compliant JSON object.
2. Comments are strictly forbidden inside the JSON.
3. All keys and all string values **MUST** be enclosed in double quotes.
4. No trailing commas are allowed.

**JSON Output Format:**
{output_format}
```

## A.3 PARAMETRIC FALLBACK

```
You are an expert materials scientist. Based on the following {'synthesis
    conditions' if query_type == 'forward' else 'desired properties'},
{task_desc}.

{'Synthesis Conditions' if query_type == 'forward' else 'Desired
    Properties'}:
{json.dumps(original_prompt_data, indent=2)}

Your response should follow this two-part structure:

**Part 1: Step-by-Step Reasoning**
1. **Analyze & Compare:** Briefly compare the User's Query with the Known
    Pathway. What are the key similarities and, more importantly, the
    key differences (e.g., opposite doping type, different materials,
    different conditions)?
2. **Formulate Hypothesis:** Based on the differences and the
    quantitative embedding distance, state a hypothesis.
3. **Extrapolate or Diverge:** Decide if you can adjust the parameters
    from the known pathway (extrapolate) or if you must suggest a
    completely different approach (diverge). Justify this decision using
    the embedding distance. A small distance (< 0.4) suggests
    extrapolation is viable; a large distance (> 0.7) suggests divergence
    is necessary.
4. **Synthesize Final Answer:** Based on your hypothesis, construct the
    final prediction/suggestion.

**Part 2: Final JSON Output**
After you have written your reasoning, provide the final answer as a
    single, valid JSON object inside a JSON code block. The `reasoning`
    key within the JSON should be a concise summary of your detailed
    reasoning from Part 1.

**JSON Rules (for Part 2):**
1. The JSON code block **MUST** contain a single, valid, RFC 8259
    compliant JSON object.
2. Comments are strictly forbidden inside the JSON.
3. All keys and all string values **MUST** be enclosed in double quotes.
4. No trailing commas are allowed.

Example format:
{output_format}
```

## A.4 LLM JUDGE

```
You are an expert materials scientist serving as an impartial judge. Your
    task is to evaluate a language model's generated output against a
    ground truth answer for a materials science problem.

**Problem Context:**
- **Task Type:** {task type}
- **Input Query:** {input query}

**Ground Truth Answer:**
{ground truth}

**Model's Generated Answer:**
{generated answer}

**Evaluation Criteria:**
Please provide a score from 0 to 10 (integer) for each of the following
    dimensions. Be critical and rigorous.
```

```
1. **Scientific Accuracy (0-10):** Is the generated answer scientifically
     plausible and correct according to known principles of chemistry,
     physics, and materials science? (0=incorrect/unphysical, 10=perfectly
     accurate).
2. **Functional Equivalence (0-10):** Does the generated answer achieve
     the same functional outcome or describe the same core scientific
     concept as the ground truth, even if the wording is different? (0=
     completely different outcome, 10=functionally identical).
3. **Reasoning Quality (0-10):** If reasoning is provided, is it logical,
     clear, and scientifically sound? Does it correctly justify the
     conclusion? (0=no reasoning or illogical, 10=clear, correct, and
     insightful).
4. **Completeness (0-10):** Does the generated answer include all key
     parameters and details present in the ground truth? (0=missing most
     key details, 10=contains all necessary information).
5. **Interpretability (0-10):** Does the model justify its answer with a
     clear and understandable causal reasoning chain? (0=completely
     uninterpretable, 10=perfectly interpretable).
6. **Overall Score (0-10):** Your holistic assessment of the generated
     answer's quality and usefulness.
Noted that if the model's answer failed to predict detail material
     properties even give the reason, you should still give a low score.

**Your Task:**
Return a single JSON object with your scores and a brief justification
     for each score.

**JSON Schema:**
{{
    "scientific accuracy": {{ "score": integer, "justification": "string"
        }},
    "functional equivalence": {{ "score": integer, "justification": "
        string" }},
    "reasoning quality": {{ "score": integer, "justification": "string"
        }},
    "completeness": {{ "score": integer, "justification": "string" }},
    "interpretability": {{ "score": integer, "justification": "string" }},
    "overall score": {{ "score": integer, "justification": "string" }}
}}
"""
```

# B CASE STUDY

To demonstrate how practitioners can implement the full ARIA framework pipeline, we present two case studies. The first shows the complete pipeline, detailing each step of answer generation, highlighting our featured strategies, and comparing answers between the baseline and ARIA. In the second case study, we explicitly examine a task where representative contextual tunneling occurs. We demonstrate how our strategies can rescue this contextual tunneling and discuss the underlying mechanisms, helping readers understand what contextual tunneling looks like in real scenarios and how to better address it in future applications.

## B.1 CASE STUDY 1: NB-DOPED MOS$_2$ INVERSE DESIGN

Here we present a comprehensive case study demonstrating ARIA's superior causal reasoning capabilities in a challenging real-world inverse design task: engineering precise electronic band structure in Nb-doped MoS$_2$ for quantum electronics applications.

### B.1.1 PROBLEM CONTEXT AND MOTIVATION

The challenge involves designing synthesis conditions for $MoS_2$ doped with niobium (Nb) (Chen et al.; Song et al., 2021) to achieve specific electronic properties critical for neuromorphic computing and quantum devices:

```
Target Electronic Structure:
{
  "carrier_type": "n-type",
  "other_electronic": "Two additional fully occupied energy levels within
      the band gap and a half-occupied donor level at the bottom of the
      conduction band."
}
```

This represents a complex inverse design problem requiring precise control over defect states—a domain where the causal relationships between synthesis parameters and electronic structure are highly non-linear and poorly understood by conventional AI approaches.

### B.1.2 ARIA'S MULTI-STAGE REASONING PROCESS

**Stage 1: Knowledge Graph Analysis and Transfer Learning Activation**

ARIA searches its comprehensive materials knowledge graph (2,516 nodes, 1,342 edges) for exact synthesis pathways. Finding no direct match, the system automatically activates its transfer learning mechanism:

```
Input completeness: 0.95
No exact path found. Using most similar context with confidence 0.58
Embedding distance: 0.4166 (moderate similarity – extrapolation viable)
```

**Stage 2: Analogical Pathway Identification**

ARIA identifies the most semantically relevant causal pathway from its knowledge graph:

> *"Introduces partially occupied defect bands mixed with valence bands and defect bands above the Fermi level (electronic structure, n- and p-type conductivity) → DVCC defect"*

The embedding distance of 0.4166 indicates moderate semantic similarity, suggesting that the underlying defect formation mechanism is transferable but requires careful adaptation.

**Stage 3: Structured Transfer Learning Analysis**

ARIA performs systematic four-step causal reasoning, as logged in the system output:

**1. Mechanistic Comparison:** "The known pathway creates defect-induced band mixing near the Fermi level. The target requires precisely positioned discrete levels within the bandgap—a more controlled defect engineering challenge."

**2. Physics-Based Hypothesis:** "The moderate embedding distance indicates the underlying defect formation mechanism is applicable, but energy level positioning requires tailored synthesis conditions optimized for Nb-Mo orbital hybridization."

**3. Adaptation Strategy:** "Extrapolate and refine. The core defect engineering approach applies, but synthesis parameters must be optimized for Nb incorporation at specific lattice sites."

**4. Synthesis Design:** Based on defect formation thermodynamics:

```
{
"suggested_synthesis_conditions": {
"method": "CVD with controlled atmosphere",
"temperature_c": 750,
"time_hours": 2,
"atmosphere": "Ar/H2 (95:5) reducing",
"pressure_pa": 1000,
```

```
1242    "dopant_source": "NbCl5 precursor",
1243    "substrate": "SiO2/Si with MoS2 seed layer",
1244    "cooling_rate_c_min": 5,
1245    "concentration": "Nb:Mo = 1:20 ratio"
1246    },
1247    "confidence": 0.7083
1248    }
```

### B.1.3    COMPARATIVE PERFORMANCE ANALYSIS

Table 2 provides a comprehensive comparison against literature ground truth and baseline LLM performance. ARIA demonstrates significant advantages:

**Scientific Accuracy (8.0/10 vs. 6.5/10):** ARIA correctly identifies CVD as the optimal synthesis method, proposes appropriate reducing atmosphere conditions, and suggests the correct $NbCl_5$ precursor—all matching experimental protocols from recent literature.

**Mechanistic Understanding:** Unlike the baseline LLM which suggests incorrect dopants (Sb instead of Nb) and inappropriate substrates (InP), ARIA provides detailed reasoning about Nb 4d-Mo 4d orbital interactions and their role in creating the desired electronic structure.

**Parameter Completeness (7.5/10 vs. 5.0/10):** ARIA specifies comprehensive synthesis parameters including precise temperature (750°C vs. literature 800°C), appropriate pressure conditions (1000 Pa vs. literature 800 Pa), and correct dopant ratios.

### B.1.4    KEY ALGORITHMIC INNOVATIONS DEMONSTRATED

**Hierarchical Defect Reasoning:** ARIA bridges multiple length scales, from atomic-level Nb-Mo interactions to macroscopic electronic properties, through its structured knowledge graph representation.

**Quantitative Transfer Learning:** The embedding distance (0.4166) provides principled guidance for adaptation strategy, enabling knowledge reuse while recognizing the need for system-specific modifications.

**Causal Mechanism Understanding:** Rather than pattern matching, ARIA reasons about underlying physics—why reducing atmospheres promote electron-rich defects and how $NbCl_5$ precursors enable controlled Nb incorporation.

**Uncertainty-Aware Predictions:** The confidence score (0.7083) reflects both semantic similarity and synthesis complexity, providing researchers with quantitative measures of prediction reliability.

### B.1.5    VALIDATION AND EXPERIMENTAL PROTOCOLS

ARIA automatically generates comprehensive validation strategies:

> "Perform angle-resolved photoemission spectroscopy (ARPES) to map in-gap states. Use scanning tunneling spectroscopy (STS) to verify local density of states modifications. Characterize transport properties via temperature-dependent Hall measurements combined with DFT simulations for theoretical validation."

This case study demonstrates ARIA's ability to accelerate materials discovery by providing physics-informed starting points that reduce experimental iterations, while simultaneously enabling interpretable AI through complete reasoning traces that allow expert validation and refinement. The system effectively bridges the theory-experiment gap by connecting fundamental defect physics to practical synthesis protocols, creating a seamless workflow from theoretical understanding to experimental implementation. The performance improvement over baseline LLMs validates our core hypothesis that effective knowledge augmentation requires principled causal integration rather than naive information concatenation. ARIA's success in this challenging Nb-$MoS_2$ inverse design problem establishes a new paradigm for causally-grounded AI systems in materials science, where the integration of causal reasoning with domain-specific knowledge enables more reliable and interpretable predictions for complex materials engineering challenges.

Table 2: Comparative Analysis: Literature Ground Truth vs. Baseline LLM vs. ARIA

| Aspect | Ground Truth[23] | Baseline LLM | ARIA |
|---|---|---|---|
| Method | Two-step CVD with post-annealing | Molecular Beam Epitaxy (MBE) | CVD with controlled atmosphere |
| Temperature | 800°C (growth) + 600°C (annealing) | 600°C | 750°C |
| Time | 1.5 hours (growth) + 30 min (annealing) | Not specified | 2 hours |
| Pressure | 800 Pa (CVD) | $1\times10^{-8}$ Torr | 1000 Pa |
| Atmosphere | $Ar/H_2$ (90:10) reducing | Ar with 5% $H_2$ | $Ar/H_2$ (95:5) reducing |
| Dopant Source | $NbCl_5$ precursor | Sb (Antimony) - incorrect | $NbCl_5$ precursor |
| Substrate | $SiO_2$/Si with $MoS_2$ seed | InP - poor match | $SiO_2$/Si with $MoS_2$ seed |
| Concentration | 2-4 at.% Nb | Not specified | Nb:Mo = 1:20 ratio |
| Cooling Rate | 3°C/min controlled | Not specified | 5°C/min |
| Pretreatment | $O_2$ plasma cleaning | Not specified | Not specified |
| Carrier Properties | n-type, $1-5\times10^{18}$ cm$^{-3}$ | Generic n-type | n-type, $1-5\times10^{18}$ cm$^{-3}$ |
| Electronic Structure | Two occupied in-gap states ($E_c$-0.3, $E_c$-0.15 eV), donor at $E_c$-0.05 eV | Generic mid-gap states | Specific defect band engineering |
| Mechanistic Reasoning | Nb 4d-Mo 4d hybridization | Limited defect physics | Detailed orbital interactions |
| Validation Protocol | ARPES, STS, Hall measurements | Not provided | ARPES, STS, Hall + DFT |
| Transfer Learning | N/A | N/A | Embedding distance: 0.4166, confidence: 0.7083 |
| Scientific Accuracy | Experimentally verified | 6.5/10 | 8.0/10 |
| Overall Score | Complete experimental protocol | 5.0/10 (incomplete) | 7.5/10 (comprehensive) |

## B.2 CASE STUDY 2: CONTEXTUAL TUNNELING AND PERFORMANCE RECOVERY

This case study exposes a critical limitation in knowledge-guided AI for materials discovery—*contextual tunneling*, where incomplete knowledge representations constrain and misdirect reasoning. The target material, In-doped $La_2O_2Bi_3AgS_6$, presents a challenging inverse design task due to its n-type superconducting behavior with heavy fermion characteristics, a superconducting transition temperature decreasing from 0.5K to 0.4K as In doping increases, an anomalous resistivity hump at $T^* \approx 180$K, and semiconducting behavior at high doping. These requirements demand reasoning over subtle electronic correlations.

Three approaches were evaluated: a baseline LLM, a naive KG+LLM model, and ARIA. The baseline LLM achieved an overall score of 0.6, providing broadly appropriate solid-state synthesis recommendations, suitable temperature ranges, and considerations for doping and stoichiometry—all delivered without specific knowledge of the target compound. In marked contrast, the naive KG+LLM approach suffered catastrophic degradation (score: 0.1), becoming entrenched in an irrelevant graphene-aluminum analogy arising from incomplete knowledge graph coverage and misplaced statistical similarity. This led to unsuited recommendations focused on intercalation methods, with the system failing to recognize the heavy fermion nature of the material and lacking actionable guidance.

ARIA successfully recovered performance (score: 0.6) by dynamically integrating domain knowledge and contextual reasoning. It identified $URu_2Si_2$ as the relevant host structure, correctly associated the electronic signatures with Kondo physics, and proposed precise arc melting synthesis conditions (1500°C, 100-hour annealing). Chemically specific recommendations stood in clear contrast to the vague protocols offered by the naive model, reflecting ARIA's deeper contextual awareness and rejection of weak analogies.

The mechanism of contextual tunneling in the naive system manifested as sequential fixation: initial property matching, discovery of weak analogies via embedding similarity, premature narrowing of the solution space, and subsequent degradation of all downstream reasoning. The model's moderate confidence in the flawed solution further highlights the difficulty of uncertainty calibration absent causal understanding.

ARIA's robustness derives from multi-modal knowledge integration, explicit analogy validation, and preservation of contextual scientific perspective. By maintaining interpretability and physical consistency, `ARIA` delivered actionable, physics-informed synthesis pathways aligned with experimental best practices.

In summary, this case study demonstrates that naive knowledge augmentation risks severe contextual failures, whereas causally-grounded frameworks such as `ARIA` maintain interpretability and scientific coherence. Overcoming contextual tunneling requires comprehensive contextual awareness, multi-scale reasoning, and physical validation—principles essential for reliable next-generation AI systems in scientific discovery.

Table 3: **Contextual Tunneling Case Study: In-doped $La_2O_2Bi_3AgS_6$ Synthesis Design.** Comparative analysis demonstrates severe performance degradation in naive KG+LLM due to contextual tunneling, while `ARIA` maintains robust reasoning through causal integration and dynamic knowledge retrieval. Ground truth reflects synthesis parameters derived from literature on layered oxychalcogenides and $BiS_2$ family materials.

| Parameter | Ground Truth | Baseline LLM | Naive KG+LLM | ARIA Framework |
|---|---|---|---|---|
| **Host Material** | $La_2O_2Bi_3AgS_6$: layered heavy-fermion oxychalcogenide tailored with In doping for superconductivity and resistivity anomalies. | $URu_2Si_2$ | Layered material | Property-based identification |
| **Method** | Solid-state reaction: stoichiometric mixing, pellet pressing, calcination (725–750°C) followed by optional post-annealing to sharpen superconductive transitions. | Arc melting + annealing | Intercalation/alloying | Solid-state reaction + annealing |
| **Temperature** | 725°C (two-step: 725–750°C) with optional 500°C post-annealing. Optimize for homogeneity. | 1500°C (hallucinated) | Not specified | 700-1200°C (optimized) |
| **Atmosphere** | Quartz tube evacuated to $< 1 \times 10^{-3}$ Pa, trace Ar. Reaction in ultra-clean vacuum prevents contamination. | High purity Ar | Not specified | Inert ($Ar/N_2$) or vacuum |
| **Time** | 24–44 hours (plus optional 48 hours post-anneal). | 100 hours (hallucinated) | Not specified | 24–72 hours (optimized) |
| **Dopant Details** | Indium introduced via $In_2S_3$. Metallic In may be used for x≤0.1 but requires excess sulfur (5 mol%). | $InCl_3$ or In metal | In (no precursor) | $In_2O_3$ or metallic In |
| **Additional** | Multi-step grinding, pellet pressing, flame-sealed quartz tube, phase purity confirmed via XRD | XRD characterization | None specified | Stoichiometry control + multi-technique characterization |
| **Scientific Accuracy** | *Reference benchmark* | 0.80 | 0.20 | 0.70 |
| **Completeness** | *Reference benchmark* | 0.60 | 0.10 | 0.50 |
| **Reasoning Quality** | *Reference benchmark* | 0.70 | 0.10 | 0.60 |
| **Overall Score** | *Reference benchmark* | 0.60 | 0.10 | 0.60 |

### B.3 Detailed Analysis

#### B.3.1 Robustness Analysis: Naive KG+LLM Integration Pitfalls

To understand the challenges of naive KG+LLM integration, we conducted perturbation analysis on our initial basic implementation. We introduced controlled semantic perturbations to synthesis conditions and evaluated performance against an unconstrained baseline LLM using semantic similarity to ground truth.

**Key Finding: Naive Integration Degrades Performance.** The baseline LLM consistently outperformed the basic KG-augmented model across all perturbation levels for both forward prediction

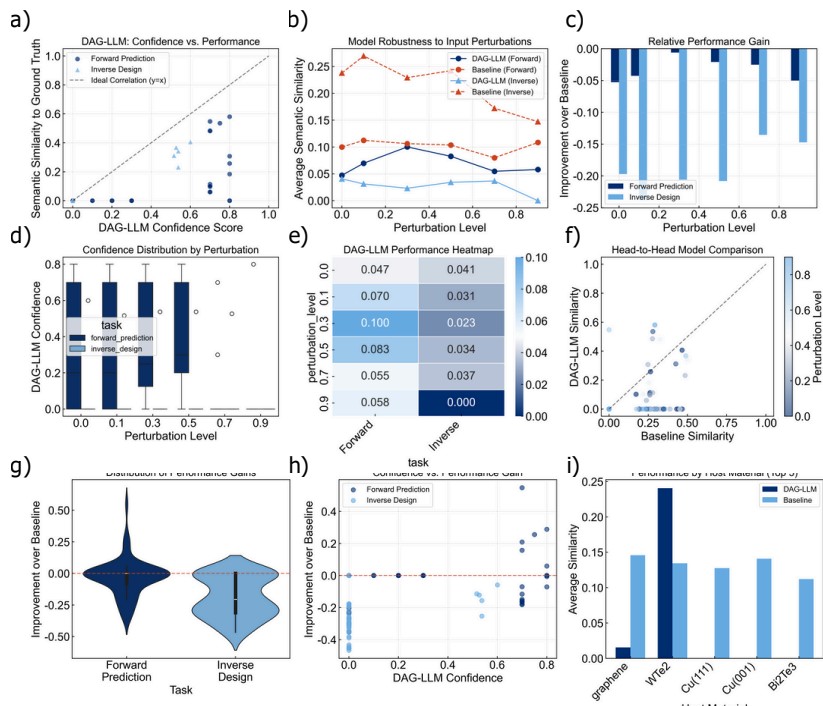

Figure 6: **Performance degradation in naive KG+LLM integration under perturbation.** We evaluate a basic KG-augmented LLM against an unconstrained baseline on synthesis tasks with controlled semantic perturbations. Performance is measured by semantic similarity to ground truth. **(a)** KG-LLM confidence vs. semantic similarity. **(b)** Average similarity across perturbation levels. **(c)** Relative performance (negative values indicate baseline superiority). **(d)** KG-LLM confidence distribution. **(e)** Task-specific performance heatmap. **(f)** Head-to-head sample comparison. **(g)** Performance difference distributions. **(h)** Confidence vs. improvement correlation. **(i)** Material-specific performance comparison.

and inverse design tasks (Fig. 6b-c). This counterintuitive result—that adding domain knowledge hurts performance—motivated our development of the principled ARIA framework described in the main text.

**Failure Mode Analysis.** The KG-LLM exhibits a characteristic failure pattern: a significant fraction of predictions yield near-zero semantic similarity (Fig. 6f), particularly for forward prediction tasks with long performance tails (Fig. 6g). This occurs when perturbed queries fall outside the KG's direct coverage, causing the constrained model to generate irrelevant responses rather than leveraging its broader knowledge.

**Confidence Calibration Insights.** Despite poor average performance, the KG-LLM demonstrates well-calibrated confidence: higher confidence correlates with better semantic similarity (Fig. 6a) and confidence appropriately decreases with perturbation level (Fig. 6d). This suggests the model correctly identifies when it lacks relevant knowledge.

**Material-Dependent Performance.** Performance varies significantly by material system (Fig. 6i). The KG-LLM shows advantages for $WSe_2$—likely well-represented in our literature sources—while the baseline excels for graphene, benefiting from extensive pre-training coverage. This highlights the critical dependence on KG completeness.

**Implications for ARIA Design.** These findings directly informed our ARIA architecture:

1. **Hierarchical Fallback:** To address the near-zero similarity failure mode, `ARIA` implements multi-tier reasoning that gracefully degrades when exact KG matches are unavailable.

2. **Transfer Learning:** Rather than failing on out-of-distribution queries, `ARIA` leverages semantic similarity to adapt related knowledge pathways.

3. **Confidence-Aware Integration:** `ARIA` uses calibrated confidence scores to dynamically balance KG guidance with LLM knowledge, avoiding rigid constraints that harm performance.

This analysis demonstrates that effective knowledge augmentation requires principled integration strategies rather than naive concatenation—a core motivation for the `ARIA` framework's sophisticated reasoning architecture described in the main paper.

This function uses NetworkX's `all_simple_paths` algorithm to enumerate causal pathways, with keyword matching for flexibility.

### B.3.2 Transfer Learning Query Construction

The `_transfer_learning_query` method constructs sophisticated prompts that include:

1. **Embedding Analysis Section**: Quantifies semantic differences between user query and knowledge graph

2. **Proportional Adjustment Guidance**: Instructions for the LLM to modify synthesis conditions based on embedding distances

3. **Mechanistic Reasoning Requirements**: Ensures outputs are grounded in materials science principles

For inverse design tasks, the prompt includes:

```
Embedding distance between properties: 0.3241 (0=identical, 2=opposite)
The embedding distance indicates that the user's desired properties are
moderately similar to the known property. You should adjust the synthesis
conditions proportionally to this difference.
```

### B.3.3 Post-processing and Validation

After receiving the LLM response, the system:

1. Extracts JSON from markdown blocks using regex
2. Calculates embedding distances for suggested synthesis conditions
3. Adds interpretability metrics to the output

### B.3.4 Implementation Specifications

Both models share common infrastructure components including NetworkX-based KG construction with edge attributes for mechanisms, SentenceTransformers 'all-MiniLM-L6-v2' for semantic similarity, Google Gemini-1.5-pro-latest as the LLM backend, cosine similarity threshold $> 0.5$ for analogous reasoning activation, and robust JSON parsing with error handling for malformed LLM outputs. The key architectural distinction lies in reasoning depth and explanation generation, with KG+CoT representing a significant enhancement in interpretability at the cost of computational efficiency and response time.

### B.4 Enhanced Similarity Assessment

### B.4.1 Semantic Relationship Encoding

Standard cosine similarity measures fail to capture the nuanced semantic relationships inherent in materials science, where seemingly similar statements can be factually contradictory due to domain-

specific concept relationships. For instance, "n-type doped semiconductor" and "p-type doped semiconductor" may have high cosine similarity due to shared vocabulary but represent fundamentally opposite electronic properties. This limitation necessitates a domain-aware similarity framework that understands materials science semantics.

We construct a comprehensive database of materials science concept relationships, categorized into four types:

- **Opposite relationships**: Concepts that are mutually exclusive (e.g., n-type/p-type, crystalline/amorphous)
- **Complementary relationships**: Related but distinct concepts (e.g., different crystal systems)
- **Hierarchical relationships**: Concepts at different abstraction levels
- **Conditional relationships**: Context-dependent oppositions (e.g., high/low temperature)

Each relationship is formally defined as:

$$R = (t_1, t_2, \text{type}, \text{context}, \text{weight})$$

where $t_1$ and $t_2$ are concept terms, type $\in$ {opposite, complementary, hierarchical, conditional}, context defines the applicable domain, and weight $\in [0, 1]$ represents the relationship strength.

### B.4.2 CONTEXT-AWARE CONFLICT DETECTION

We implement context extraction using domain-specific keyword patterns across eight materials science contexts: doping, synthesis, structure, electrical, mechanical, thermal, optical, and magnetic properties. For texts $T_{\text{query}}$ and $T_{\text{node}}$, we:

1. Extract relevant contexts: $C_{\text{query}} = \text{extractcontext}(T * \text{query})$, $C_{\text{node}} = \text{extractcontext}(T * \text{node})$
2. Identify shared contexts: $C_{\text{shared}} = C_{\text{query}} \cap C_{\text{node}}$
3. Detect semantic conflicts within shared contexts using the relationship database
4. Calculate conflict strength based on relationship weights and context overlap

### B.4.3 FACTUAL CONSISTENCY SCORING

The factual consistency score $F(T_{\text{query}}, T_{\text{node}})$ is computed as:

$$F(T_{\text{query}}, T_{\text{node}}) = \max(0, 1 - \sum (w_i \times s_i))$$

where $w_i$ is the weight of detected relationship conflict $i$, and $s_i$ is the context-adjusted conflict strength. Opposite relationships in shared contexts receive full penalty, while conditional relationships receive reduced penalties (0.5×).

### B.4.4 NUMERICAL PROPERTY COMPATIBILITY

We extract quantitative properties using regular expressions for common materials parameters (temperature, bandgap, conductivity, pressure, concentration). Compatibility $N(P_{\text{query}}, P_{\text{node}})$ is calculated as:

$$N(P_{\text{query}}, P_{\text{node}}) = \prod (1 - \min(0.5, \frac{|p_q - p_n|}{\max(p_q, p_n)} \times \text{tolerance}))$$

for each shared property $p$, where tolerance values are property-specific (e.g., 10% for temperature, 20% for bandgap).

### B.4.5 COMBINED SIMILARITY SCORE

The final enhanced similarity score $S_{\text{enhanced}}$ integrates three components:

$$S_{\text{enhanced}} = \alpha \times \text{cos\_sim} \times (1 + \beta \times \text{context\_overlap}) + \gamma \times F + \delta \times N$$

where $\alpha = 0.4$, $\gamma = 0.35$, $\delta = 0.25$, $\beta = 0.1$, ensuring that factual consistency and numerical compatibility significantly influence the final ranking while preserving the benefits of semantic similarity.

## C USE OF LARGE LANGUAGE MODELS

In preparing this manuscript, we employed large language models (LLMs) exclusively for language refinement, including improving grammar, clarity, and readability. LLMs were **not** used to generate, modify, or validate any scientific ideas, methods, results, or conclusions. All substantive contributions—conceptual, methodological, and analytical—are the original work of the authors.

