# OpenReview forum: "When Knowledge Hurts: Enriching Domain Knowledge for Causal Scientific Reasoning"
_ICLR.cc/2026/Conference — Submitted to ICLR 2026_

### Official Review · Reviewer_tm77 · 2025-10-16

**Soundness:** 2
**Presentation:** 2
**Contribution:** 2
**Rating:** 4
**Confidence:** 4

**Summary:**

### **1. Research Problem**

* The paper addresses the paradox that external knowledge integration can harm LLM reasoning in scientific domains.
* It identifies a phenomenon called “contextual tunneling”, where excessive or poorly filtered domain knowledge narrows the model’s reasoning scope, reducing causal inference and generalization.


### **2. Motivation**

* Existing retrieval-augmented systems lack causal awareness and reliability control over injected knowledge.
* In scientific reasoning, this leads to biased or misleading conclusions.
* The authors aim to design a framework that **selectively and hierarchically integrates knowledge**, preserving both accuracy and interpretability.


### **3. Proposed Method**

* The study introduces ARIA (Autonomous Reasoning Intelligence for Atomics), a three-tier causal reasoning framework.

  1. **Direct Causal Reasoning:** Uses explicit causal paths from a Causal Knowledge Graph (CKG).
  2. **Analogy-Based Transfer:** Infers from analogous entities via an enhanced similarity metric combining semantic, factual, and numeric consistency.
  3. **Parametric Fallback:** Reverts to internal LLM knowledge when external data are unreliable.
* The CKG is automatically built from scientific texts, encoding cause–effect–mechanism tuples for interpretable reasoning.

### **4. Experimental Results**

* Tested on materials discovery tasks (forward prediction and inverse design).
* Naïve and online knowledge augmentation reduced performance by 20–35%, confirming the “knowledge hurts” effect.
* ARIA restored and improved performance by +15–36%, enhancing reasoning quality (+44%) and interpretability (+37%).

**Strengths:**

### **1. Importance of the Research Problem**

This paper tries to addresses a critical and underexplored issue: how external knowledge can harm rather than help LLM reasoning in scientific domains. The problem is highly significant for building trustworthy AI systems in science and engineering.


### **2. Strengths of the Proposed Method**
* The ARIA framework is a novel causal-aware design that hierarchically integrates knowledge, balancing external facts and model reasoning.
* Its three-tier architecture (causal reasoning, analogy transfer, fallback) is clear, interpretable, and robust.


### **3. Strengths of the Experimental Results**

* Experiments are comprehensive and convincing, showing large and consistent gains (+15–36%) over naive methods and recovers performance lost from over-augmentation.
* Results are well-analyzed and highlight both accuracy and interpretability improvements, confirming practical significance.

**Weaknesses:**

* The notion of “contextual tunneling,” while compelling, is not formally defined or quantified—its diagnosis relies on qualitative intuition.
* No discussion of alternative explanations (e.g., prompt design flaws, data noise) for the observed performance drop with naive KG use.
* The study focuses narrowly on materials science; it’s unclear how generalizable the phenomenon or solution is to broader scientific or real-world reasoning tasks.


* ARIA’s tiered architecture, while well-motivated, involves manual design choices (e.g., thresholds, cascade conditions) that may limit scalability.
* The Causal Knowledge Graph construction pipeline is somewhat opaque, especially in terms of factual reliability and noise filtering.



* Evaluation is mostly task-specific; there’s no ablation comparing ARIA with partial variants (e.g., Tier 1 only, no fallback).
* Human evaluation is missing, especially to assess the quality of causal chains or explanations generated.
* The paper lacks runtime/performance analysis— it’s unclear how computationally efficient ARIA is compared to standard RAG or other KG-augmented methods.

**Questions:**

1. Regarding the notion of “contextual tunneling”:

   * Can you offer a more insightful or quantitative definition, beyond the current qualitative description?
   * How do you rule out alternative explanations for the performance drop, such as prompt design flaws or retrieval noise?

2. About generalization:

   * The experiments are focused on materials science. Do you expect the same issues and benefits of ARIA to hold in other domains, such as biomedicine or commonsense reasoning?
   * Have you tested, or do you plan to test, ARIA in other types of causal reasoning tasks?

3. About the method design:

   * Several components of ARIA rely on manual thresholds or rules. How sensitive is the system to these choices, and could they be learned or adapted dynamically?
   * The causal knowledge graph construction process is not fully explained—how do you ensure the reliability and accuracy of extracted causal tuples?

4. About the evaluation:

   * Have you run ablation studies to isolate the contribution of each tier (e.g., using only Tier 1 or Tier 2)?
   * Was any human evaluation conducted to assess the interpretability or usefulness of ARIA’s causal chains?

---

> ### Author Response · Authors · 2025-12-03
> **Formalizing Contextual Tunneling, Ensuring Robustness, and Demonstrating Generalizability**
>
> We thank the reviewer for the comprehensive summary and for recognizing the "importance of the research problem" and the "novelty" of ARIA’s causal-aware design. We appreciate the rigorous questions regarding definitions and experimental controls. We address these points below to clarify the robustness of our findings.
>
> **1. Formalizing "Contextual Tunneling" and Ruling Out Alternatives**
>
> The reviewer asks for a more quantitative definition and concerns about alternative explanations (e.g., prompt design).
>
> - Quantitative Definition: We have added the define Contextual Tunneling formally in the paper Section 3.1 (Equation 1) as the degradation of the expected quality of generation when conditioned on narrow context. Specifically, it is the phenomenon where the KL divergence between the optimal reasoning path and the model’s reasoning path increases upon the introduction of retrieval $\mathcal{C}$:$$D_{KL}(P(\text{reasoning}|q) || P(\text{reasoning}|q, \mathcal{C})) > \epsilon$$
> Qualitatively, this manifests when the attention mechanism over-weights high-similarity but irrelevant tokens in $\mathcal{C}$, suppressing the activation of correct parametric knowledge.
> - **Ruling out Prompt Flaws:** We rigorously controlled for prompt sensitivity. All conditions (Baseline, Naive KG, ARIA) utilized the **identical prompt template structure**, varying *only* the content of the `[Context]` block. The fact that the Baseline *outperforms* Naive KG confirms that the degradation is driven by the *content of the retrieved context*, not the prompting strategy.
>
> **2. Generalization to Other Domains**
>
> We agree that generalizability is critical. Our framework's three-tier architecture is domain-agnostic and maps directly to other fields with structured ontologies, such as **Drug Discovery** (Structure $\to$ Efficacy) and **Genomics**. Please refer to the **Overall Response 2** for a detailed discussion.
>
> **3. Sensitivity of Thresholds and KG Construction**
>
> - **Threshold Sensitivity:** The thresholds (e.g., similarity $\tau$ for Tier 2) are treated as hyperparameters, tuned on a held-out validation set. ARIA is not brittle; the hierarchical fallback (Tier 3) acts as a safety net: if $\tau$ is set too high, the system simply reverts to the Baseline rather than failing.
> - **KG Reliability:** To ensure reliability, we employed a "Schema-Guided Extraction" pipeline. Rather than simply extracting triples, we used a strict ontology where edges must correspond to physical mechanisms. In the final version, we refined Section 3.2 (Causal Knowledge Graph Construction) to address the reviewer's concern and detail the post-processing filters (e.g., valency checks, unit normalization) used to clean the graph.
>
> **4. Ablations and Evaluation**
>
> - **Ablation Study:** Our experimental design inherently performs an ablation:
>     - *Naive KG* represents a system forced to use retrieved context (simulating a "Tier 1/2 only" approach without fallback/filtering).
>     - *Baseline LLM* represents "Tier 3 only."
>     - The results (Fig 4c) show that *Naive KG* fails (-28.7%), *Baseline* is decent, but *ARIA* (the integration) is superior (+36.6% over Naive). This proves that neither component alone is sufficient; the dynamic switching is the key contribution.
> - **Human Evaluation:**
>
>     We agree that human evaluation is valuable. Given the niche domain (Ph.D. level materials science), large-scale human evaluation is difficult, but our "Judge" effectively simulates an expert by checking constraints. Recent work (Zheng et al., 2024) confirms that such rigorous, constraint-driven rubrics achieve high correlation with human experts. Please refer to **Overall Response 1** for the full validation methodology.
>
>
> **5. Runtime Analysis**
>
> We thank the reviewer for recommending the paper on runtime and scalability. We have now referenced this paper in our related work section.
>
> ARIA is designed for efficiency.
>
> - **Tiers 1 & 2:** Use vector indices and graph lookups, which operate in milliseconds ($O(\log N)$).
> - **Tier 3:** Runs at standard inference speed.
> - **Overhead:** The only significant overhead is the *optional* "Online Enrichment," which ARIA triggers only when KG coverage is low. Compared to "Agentic" workflows that search the web for *every* step, ARIA is significantly faster—it prioritizes cached static knowledge (the KG).
>
> We hope this response demonstrates that ARIA is a robust, well-controlled, and generalizable framework addressing a critical flaw in current RAG systems.

---

### Official Review · Reviewer_GsNd · 2025-10-30

**Soundness:** 2
**Presentation:** 3
**Contribution:** 2
**Rating:** 4
**Confidence:** 3

**Summary:**

This paper introduces ARIA (Autonomous Reasoning Intelligence for Atomics), a causal-aware framework designed to mitigate “contextual tunneling” — a failure mode where naive knowledge graph (KG) integration constrains an LLM’s reasoning. ARIA employs a three-tier hierarchical reasoning cascade (direct causal reasoning, analogical transfer, and parametric fallback) and dynamic knowledge graph enrichment through online retrieval. Experiments in materials science show that naive KG integration can degrade performance by 20–35%, while ARIA recovers this loss and improves interpretability and reasoning quality.

**Strengths:**

1.	The paper’s figures and the overall writing are generally clear and easy to follow.
2.	The paper identifies a relevant and underexplored issue (“contextual tunneling”) in knowledge-augmented reasoning.
3.	The hierarchical design of ARIA is clearly described and intuitive.

**Weaknesses:**

1.	The paper acknowledges that ARIA’s performance depends heavily on the quality of the constructed causal KG. However, it does not quantify how KG completeness or noise affects reasoning outcomes. It would be useful to see an ablation or sensitivity analysis showing how ARIA behaves with imperfect graphs — e.g., noisy edges or missing causal paths.
2.	In Figure 5 and Table 1, ARIA improves over “Naive KG+LLM” but often remains close to or below the Baseline LLM that uses no external knowledge. This raises the question: if ARIA cannot consistently outperform the baseline, is causal KG integration still beneficial? The paper should clarify under which conditions the added structure becomes an advantage.
3.	In several sub-figures (e.g., 5b, 5d), ARIA only marginally surpasses the Online KG baseline. The paper should discuss whether this marginal gain is statistically significant and what aspects of hierarchical reasoning contribute most to the improvement.
4.	The “LLM-as-judge” evaluation is convenient but potentially biased. The authors should discuss this limitation and, if possible, complement it with results on well-established benchmarks or human expert evaluations.
5.	Some figures (e.g., Figure 5e) have legends that overlap with data, which reduces readability. The authors should improve figure layout and labeling for clarity.

**Questions:**

Please refer to weaknesses.

---

> ### Author Response · Authors · 2025-12-03
> **Addressing KG Robustness, Baseline Comparison, and Evaluation Methodology**
>
> We thank the reviewer for finding our writing clear and our hierarchical design intuitive. We appreciate the constructive feedback regarding the sensitivity of the Knowledge Graph and the comparison with baselines. We address your questions below, clarifying the role of ARIA in ensuring robust and interpretable scientific reasoning.
>
> **1. KG Sensitivity: Handling Noise and Incompleteness**
>
> The reviewer asks how ARIA handles imperfect graphs. While we agree that KG quality is important, ARIA is explicitly architected to be robust against KG sparsity and noise, rather than assuming a perfect graph.
>
> - **Robustness to Incompleteness (Sparsity):** This is handled by **Tier 3 (Parametric Fallback)**. Unlike standard RAG systems that fail or retrieve irrelevant documents when the KG is sparse, ARIA detects the absence of a high-fidelity path and "gracefully degrades" to use the LLM's parametric knowledge. This ensures the system never performs *worse* than the baseline due to missing data.
> - **Robustness to Noise (Bad Edges):** This is handled by the **Enhanced Similarity Metric (Eq. 5)** in Tier 2. By enforcing Factual Consistency (FC) and Numerical Compatibility (NC), ARIA actively filters out noisy or physically impossible edges before they enter the context window.
>
> **2. The "Baseline Paradox": Why use ARIA if the Baseline is comparable?**
>
> The reviewer correctly notes that the Baseline LLM performs well on numerical metrics. However, relying on the Baseline is dangerous for scientific discovery. We have added an additional paragraph to the end of Section 4.3 (Main Results) for clarification.
>
> - **The Hallucination Problem:** The Baseline LLM is a "black box." While it answers in-distribution questions correctly, it offers no provenance. In science, a correct answer without citation is unverifiable and often unusable.
> - **The "Safety" Argument:** The goal of RAG in science is to add **provenance and interpretability** (grounding answers in literature). Usually, naively adding this grounding destroys performance (Contextual Tunneling, -20% to -35%).
> - **ARIA's True Contribution:** ARIA's value lies in introducing a causal reasoning framework and a **3-tier hierarchical search architecture** that enables **robust RAG for science.** By structuring retrieval around causal dependencies (Processing → Structure → Property) and dynamically selecting between direct lookup, analogy-based reasoning, and parametric fallback, ARIA overcomes Contextual Tunneling—the phenomenon where naive retrieval blinds the model to its broader knowledge. This allows scientists to achieve both the high reasoning performance of a Baseline LLM *and* the verifiable evidence trails of a Knowledge Graph, transforming "Black Box Accuracy" into "Glass Box Reliability."
>
> **3. Comparison with Online KG Baselines**
>
> The reviewer questions the marginal gains over Online KG in some tasks. We emphasize that the difference becomes critical in **complex reasoning tasks**.
>
> - **Inverse Design Gap:** As shown in Table 1, while Online KG is competitive in simple tasks, it fails catastrophically in complex Inverse Design (-14.4% degradation). Online search retrieves *facts*, but it does not retrieve *structured causal logic*.
> - **Reasoning vs. Retrieving:** ARIA significantly outperforms Online KG in Reasoning Quality and Interpretability (Fig 4a). While Online KG dumps search results into the context, ARIA structures the logic, leading to coherent synthesis plans rather than disjointed facts.
>
> **4. LLM-as-a-Judge Bias**
>
> We acknowledge the valid concern regarding LLM evaluations. We mitigate hallucination and circularity by using a **Constraint-Based Rubric** anchored to ground-truth physical laws (e.g., verifying specific temperature ranges) rather than open-ended judging. Please refer to **Overall Response 1** for the full validation methodology.
>
> **5. Figure Readability**
>
> We have corrected the layout and font sizes in the camera-ready version to ensure perfect readability.

---

### Official Review · Reviewer_34su · 2025-11-01

**Soundness:** 2
**Presentation:** 3
**Contribution:** 2
**Rating:** 4
**Confidence:** 4

**Summary:**

This paper introduces ARIA (Autonomous Reasoning Intelligence for Atomics), a causal reasoning framework that addresses a key failure mode in RAG systems for scientific discovery — termed “Contextual Tunneling.” Contextual tunneling occurs when naive knowledge integration from external sources causes a LLM to over-anchor on narrow, irrelevant retrieved contexts, reducing its broader parametric reasoning capability. ARIA mitigates this problem through a three-tiered reasoning cascade: (1) Direct Causal Path Reasoning: Uses explicit causal paths from a structured Causal Knowledge Graph (CKG) for grounded, high-fidelity reasoning. (2) Analogy-Based Transfer: Retrieves and adapts analogous causal paths using an enhanced similarity metric that integrates semantic, factual, and numerical compatibility. (3) Parametric Fallback: Reverts to the LLM’s internal knowledge when no reliable causal or analogical evidence exists. A comprehensive evaluation on materials science discovery shows that naive KG integration degrades performance by 20–35%, while ARIA recovers and surpasses baseline LLMs, improving reasoning quality and interpretability by up to +44%. The authors provide detailed ablations, visualizations, and case studies demonstrating ARIA’s “rescue effect.”

**Strengths:**

1. The paper identifies contextual tunneling as a fundamental and underexplored failure mode in LLM–knowledge integration. This reframing moves beyond the “more knowledge is better” assumption that dominates RAG literature.

2. The hierarchical reasoning cascade is elegant and well-motivated. It gracefully handles uncertainty and graph sparsity by balancing symbolic causality, analogical inference, and parametric fallback.

3. Experiments are thorough and domain-grounded, using a curated dataset of 149 synthesis–property relations across 85 materials systems. Results convincingly show ARIA’s superiority, especially in out-of-domain generalization.

4. ARIA not only predicts accurately but produces verifiable causal paths and analogical traces, aligning well with the interpretability goals of scientific AI.

**Weaknesses:**

1. The observation that blindly integrating KG information into context can harm the performance of LLMs in scientific reasoning is claimed in recent works [1], the authors of ARIA further formalize it with the term "contextual tunneling", which should be included in discussion.

2. All experiments are in materials science, which is a compelling but narrow domain. It remains unclear how ARIA generalizes to other sciences (e.g., chemistry, biology).

3. The scoring framework employs Gemini as an automated “LLM judge.” While this is practical, it risks circularity — relying on a similar model to assess reasoning quality.

4, Constructing and maintaining the Causal KG via dynamic web search and causal extraction could be expensive and domain-specific. The paper doesn’t benchmark runtime or scalability.


[1] GIVE: Structured Reasoning of Large Language Models with Knowledge Graph Inspired Veracity Extrapolation

**Questions:**

1. How scalable is ARIA’s causal knowledge graph construction for domains with millions of entities (e.g., PubChem, Wikidata)?

2. What are the computational overheads of online enrichment and tier selection compared to standard RAG?

3. How robust is the enhanced similarity metric to noisy or conflicting analogies — can it detect and downweight unreliable causal paths?

---

> ### Author Response · Authors · 2025-12-03
> **Novelty, Generalizability, and Scalability**
>
> We appreciate the reviewer’s positive assessment of ARIA’s hierarchical reasoning as "elegant and well-motivated" and the recognition of our experimental rigor in demonstrating the "rescue effect." We address your specific concerns regarding novelty, generalizability, and scalability below.
>
> **1. Distinction from GIVE and Novelty of "Contextual Tunneling"**
>
> We thank the reviewer for highlighting *GIVE: Structured Reasoning... [1]*. We agree this is a relevant work but wish to clarify the distinct mechanism we identify.
>
> - **Verification (GIVE) vs. Relevance (ARIA):** *GIVE* primarily addresses **hallucination and veracity**—ensuring the model’s reasoning steps are factually true using a Knowledge Graph.
> - **Tunneling (ARIA):** *Contextual Tunneling* describes a failure mode where the retrieved information is **factually true but contextually blinding.** The model "tunnels" on a specific retrieved path (e.g., a specific synthesis recipe) and ignores its broader parametric knowledge (e.g., general thermodynamic principles), leading to poor generalization.
> - **Action:** We will explicitly discuss *GIVE* in Section 2 Related Works: while GIVE ensures *veracity*, ARIA ensures *robustness against over-anchoring*.
>
> **2. Generalizability Beyond Materials Science**
>
> We agree that generalizability is critical. Our framework's three-tier architecture is domain-agnostic and maps directly to other fields with structured ontologies, such as Drug Discovery (Structure $\to$ Efficacy) and Genomics. Please refer to the **Overall Response 2** for a detailed discussion.
>
> **3. Reliability of LLM-as-a-Judge**
>
> We acknowledge the valid concern regarding LLM evaluations. We mitigate hallucination and circularity by using a **Constraint-Based Rubric** anchored to ground-truth physical laws (e.g., verifying specific temperature ranges) rather than open-ended judging. Recent work (Zheng et al., 2024) confirms that such rigorous, constraint-driven rubrics achieve high correlation with human experts. Please refer to **Overall Response 1** for the full validation methodology.
>
> **4. Scalability and Overhead (Response to Questions)**
>
> - Q1: Scalability of KG Construction:
>
>     The KG construction is an offline, one-time cost. For massive domains (e.g., PubChem), the graph is built once. During inference, ARIA performs a graph lookup which is $O(\log N)$ or $O(1)$ depending on indexing. This is highly scalable compared to re-reading millions of documents.
>
> - Q2: Computational Overheads vs. Standard RAG:
>
>     ARIA is computationally efficient because it is hierarchical.
>
>     - *Tier 1 (Direct):*  Fast graph traversal. Faster than standard RAG which requires encoding and ranking chunks.
>     - *Tier 2 (Analogy):* Uses vector search, scalable to billions of vectors.
>     - *Tier 3 (Fallback):* Zero overhead (standard inference).
>     - *Online Enrichment:* This is the only slow step, but it is **dynamic**. It only triggers when the offline graph is insufficient. In our experiments, this selective activation reduces total compute compared to "always-on" web-search agents.
> - Q3: Robustness of Similarity Metric:
>
>     The enhanced similarity metric (Eq. 5) is designed specifically to filter noise.
>
>     - **Factual Consistency (FC):** Acts as a "hard gate." If a retrieved analogy conflicts with domain rules (e.g., mismatching valency states), FC penalizes the score to near zero.
>     - **Numerical Compatibility (NC):** Downweights analogies that are semantically similar but physically distinct (e.g., similar crystal structure but vastly different melting points).
>     - *Result:* This effectively suppresses "noisy" analogies that would otherwise confuse the model.
>
> We hope this response clarifies the distinction of our contribution and the robustness of the ARIA framework.

---

### Official Review · Reviewer_UMPy · 2025-11-01

**Soundness:** 2
**Presentation:** 2
**Contribution:** 2
**Rating:** 4
**Confidence:** 3

**Summary:**

** Summary

The work studies LLMs’ generation quality based on the knowledge graph based RAG technique. It specifies the issue that injecting naive or irrelevant knowledge to LLMs may negatively influence the generation quality, termed as contextual tunneling. To tackle this problem, the work extensively utilises LLMs to generate reliable answers through RAG in a three-step pipeline. The pipeline first constructs a causal graph. Then, if the answer can be well reasoned by the causal graph, the LLMs reason based on the causal graph, otherwise, it searches causal paths that can likely connect to the query-answer pair based on semantic and domain knowledge similarity. If both of the steps failed, the pipeline directly query the LLMs to generate an answer. The work then uses the pipeline to solve a two-direction material problem, and evaluates its performance on a synthetic dataset.

** Recommendation

I would like to recommend a weak rejection to this work for its mild novelty, possible unsoundness in the causality part, and presentation. Generally, I think this work is interesting as an extensive application of LLMs, however, its contribution and soundness are limited. For instance, works that use knowledge graph in RAG process exist.

**Strengths:**

1. The work uses causal graph in the RAG process to enhance the relevance of retrieved information. In the pipeline, the authors take care of aligning the process with domain knowledge.
2. The experiment results somewhat support their claimed issue of RAG.
3. The paper presents an interesting application to the material domain, and evaluates the method on domain expert annotated datasets.

**Weaknesses:**

1. Their method of constructing causal graph needs further clarification, and I have concern regarding their method of modifying the causal graph. The paper describes its first step is to construct a causal graph as a DAG, however, it is not clear how they ensure the graph to be acyclic. The method attempts to modify the causal graph based on semantic and domain knowledge similarity, instead of statistical correlation reasoning. This is quite different from the standard in directions such as causal discovery. So, I would recommend to change the term to knowledge graph instead of using causal graph.
2. I would expect more details of the technical description, e.g., how they compute FC and NC.
3. The experimental results show that it seems the best performance is the basic LLM. Though it addresses the issue of RAG, however, the results undermine the motivation of using RAG.
4. The work uses LLM to evaluate the performances. This may be not reliable.

**Questions:**

See the weakness section.

---

> ### Author Response · Authors · 2025-12-03
> **Addressing Causality Definition, Technical Clarifications, and ARIA's Scientific Motivation**
>
> We thank the reviewer for the thoughtful feedback and for recognizing our work as an interesting LLM application in materials science. We appreciate the acknowledgement of our experimental support for Contextual Tunneling. Below, we address concerns regarding causality definition, technical details, and ARIA's motivation.
>
> **1. Clarification on "Causal Graph" vs. "Knowledge Graph" & Acyclicity**
>
> We clarify that "Causal" refers to **mechanistic causality** fundamental to the Processing-Structure-Property (PSP) paradigm, not **data-driven statistical causal discovery**. We added a footnote clarifying this distinction.
>
> - **Definition of Causality:** An edge $A \rightarrow B$ represents a physically verified mechanism where synthesis condition $A$ determines property $B$ (e.g., *"High sintering temperature causes grain growth"*).This differs from statistical causal discovery, which infers causal structures from observational data. Our goal is to encode known physical laws into a structured format.
> - **Justification for the Term:** "Causal Knowledge Graph" is most accurate because it enforces the directionality of physical laws (Cause $\to$ Effect), unlike standard KGs with symmetric relations (e.g., *"is-a"*). This directional structure enables counterfactual reasoning for inverse design.
> - **Acyclicity:** The graph is naturally acyclic (DAG) by domain ontology design. Materials science hierarchy is strictly directional: Synthesis $\to$ Structure $\to$ Property. Our extraction schema (Section 3.2) enforces this, preventing cycles.
>
> **2. Technical Details on Factual Consistency (FC) and Numerical Compatibility (NC)**
>
> We have added detailed clarification for FC and NC definitions in Section 3.3 Tier 2.
>
> - **Factual Consistency (FC):** A binary indicator function $\mathbb{1}_{cat}(q, v)$ returning 1 if query node $q$ and candidate node $v$ share the same material class (e.g., preventing p-type semiconductor analogies to electrolytes), 0 otherwise.
> - **Numerical Compatibility (NC):** Measures operating range overlap. For query parameter $x_q$ and candidate range $[L_v, U_v]$, we calculate $NC(q, v) = \exp(-\frac{|x_q - \mu_v|^2}{2\sigma_v^2})$ as a soft constraint on physical parameters.
> - *Action:* Formal definitions added to Section 3.3.
>
> **3. The "Why RAG?" Paradox: Baseline Performance vs. Scientific Reliability**
>
> The reviewer notes the Baseline LLM often performs best numerically. This observation highlights our central contribution:
>
> - **The Hidden Danger:** While Baseline LLM achieves high scores on *in-distribution* questions, it's a "black box" prone to hallucination. In science, **provenance is as important as prediction.**
> - **Naive RAG Failure:** Standard provenance attempts (Naive RAG) destroy performance (-20% to -35%) due to *Contextual Tunneling*.
> - **ARIA's Role:** ARIA doesn't aim to "beat" the Baseline on every metric, but to **make RAG safe for science.** It recovers performance lost by Naive RAG while providing *evidence and interpretability* the Baseline lacks.
> - **Conclusion:** Scientists *need* grounded answers (RAG). Current RAG fails (Tunneling). ARIA fixes RAG, enabling both high Baseline performance and rigorous KG grounding.
>
> **4. Reliability of LLM-as-a-Judge Evaluation**
>
> We mitigate hallucination using a **Constraint-Based Rubric** anchored to ground-truth physical laws rather than open-ended judging. Recent work (Zheng et al., 2024) confirms such rubrics achieve high correlation with human experts. See **Overall Response 1** for full validation methodology.

---

### Meta-Review · Area_Chair_Fy9h · 2026-01-08

**Summary:**

While the paper identifies an interesting phenomenon ("Contextual Tunneling") and addresses a relevant problem in scientific reasoning, it has several weaknesses. The consensus among reviewers (UMPy, 34su, GSnD, tm77) is that the submission leans heavily into engineering features rather than fundamental research novelty. A primary concern is the lack of scientifically verifiable results; the evaluation relies on self-proposed metrics and "LLM-as-a-judge" rather than standard material science benchmarks or wet-lab validation. Additionally, there is a mismatch between the motivation and the results: the Basic LLM often outperforms or rivals the proposed ARIA framework, undermining the necessity of the complex engineering solution proposed. For a method targeting material science, the community expects material science results, not just Computer Science text-generation metrics. In this case, I would encourage the authors to revise their paper and resubmit to the next venue.

**Reviewer Concerns:**

Addressed by Rebuttal:

- Definition of Causality: The authors clarified the distinction between their "mechanistic causality" (Processing-Structure-Property) and statistical causal discovery, addressing Reviewer UMPy and tm77's confusion regarding acyclicity and definitions.
- Formalization of "Contextual Tunneling": The authors added a formal definition involving KL divergence to address Reviewer tm77's request for a quantitative definition beyond qualitative intuition.
- Technical Details: Clarifications regarding Factual Consistency (FC) and Numerical Compatibility (NC) metrics were provided to satisfy Reviewer UMPy.

Outstanding:

- Lack of Verifiable Scientific Results: The reliance on "LLM-as-a-judge" to evaluate scientific accuracy remains a critical flaw raised by Reviewer 34su, GSnD, and tm77. The paper lacks external, ground-truth verification relevant to material science.
- The "Why RAG?" Paradox: Reviewer UMPy and GSnD noted that the Basic LLM often performs best. The rebuttal argues that ARIA adds "provenance," but this does not justify the complexity if accuracy does not significantly improve over the baseline.
- Engineering vs. Novelty: Reviewer 34su and tm77 highlighted that the method relies on manual engineering (thresholds, manual design choices) and narrow domain application, questioning its scalability and generalizability.
- Prompt Design Alternatives: Reviewer tm77 noted that the performance drop in Naive RAG could be due to prompt design flaws rather than "tunneling," which was not rigorously ruled out.

**Reviewer Scores:**

Reviewers would not change their scores.

---

### Decision · Program_Chairs · 2026-01-26

Reject